# Allogeneic Hematopoietic Cell Transplant for B-Cell Lymphomas in the Era of Novel Cellular Therapies: Experience from a Tertiary Canadian Center

**DOI:** 10.3390/curroncol32050285

**Published:** 2025-05-20

**Authors:** Mathias Castonguay, Jean Roy, Jean-Sébastien Claveau, Sylvie Lachance, Jean-Sébastien Delisle, Thomas Kiss, Sandra Cohen, Isabelle Fleury, Luigina Mollica, Imran Ahmad, Nadia Bambace, Léa Bernard, Denis-Claude Roy, Guy Sauvageau, Olivier Veilleux

**Affiliations:** Hematology-Oncology and Cell Therapy University Institute, Hôpital Maisonneuve-Rosemont Research Center, Université de Montréal, Montréal, QC H1T 2M4, Canada; mathias.castonguay@umontreal.ca (M.C.);

**Keywords:** B-cell lymphoma, allogeneic hematopoietic cell transplant, cell therapy

## Abstract

Background: Allogeneic hematopoietic cell transplant (alloHCT) is a curative option for relapsed/refractory B-cell lymphomas (BCLs), but its role in the evolving field of cellular therapy is increasingly unclear as recent advances in transplant procedures have improved outcomes. Methods: This retrospective, single-center study included 55 BCL patients (large B-cell lymphoma—LBCL; indolent BCL; and mantle cell lymphoma—MCL) treated with alloHCT from 2015 to 2023 at Hôpital Maisonneuve-Rosemont. Primary endpoints were overall survival (OS) and progression-free survival (PFS); secondary endpoints included NRM and GVHD incidence. Results: A total of 55 patients were included (25 LBCLs, 16 indolent BCLs, 14 MCLs), and 76% of LBCLs were of indolent origin (Richter transformation, transformed follicular lymphoma). After a median follow-up of 6.1, 5.8 and 2.4 years for LBCLs, indolent BCLs and MCLs, their 5-year PFS and OS were 57.2% (IC 95%: 34.2–74.7) and 62.8% (IC 95%: 37.9–80.0), 81.2% (IC 95%: 52.5–93.5) and 93.8% (IC 95%: 63.2–99.1), and 39.0% (IC 95%: 14.3–63.3) and 68.1% (IC 95%: 35.4–86.8), respectively. The 5-year NRM was 16.9% (IC 95%: 8.2–28.3) with a relapse incidence of 23.4%. Overall/grade 3–4 acute GVHD occurred in 43.6% and 18.1% of patients. At 3 years, overall/moderate or severe chronic GVHD incidence was 49% and 34.5%. Conclusions: AlloHCT remains a potentially curative option and should be considered for fit patients with chemosensitive FL or LBCLs of indolent origin and a low comorbidity index.

## 1. Introduction

Allogeneic hematopoietic cell transplant (alloHCT) can be curative for B-cell lymphomas (BCL), but responses are heterogenous across histological subtypes. Non-relapse mortality (NRM), mainly due to graft-vs-host-disease (GVHD) and infections, remains a major concern for patients undergoing transplant. As a result, alloHCT for BCLs has been contemplated late in the course of patients with multiple relapses. However, recent changes in allo transplant strategies such as GVHD prophylaxis and treatments have changed the field and made this procedure increasingly safer. As a result, the role of alloHCT for the treatment of relapsing-refractory B-cell lymphomas (BCLs) is unclear in the rapidly evolving era of novel cellular immunotherapies and new small molecules. In this study, we aimed to examine both outcomes and toxicity of BCLs after alloHCT with the goal of better identifying which sub-types of patients could specifically benefit from this therapy.

## 2. Materials and Methods

This is a retrospective, unicentric study evaluating the outcomes of BCL patients treated with alloHCT, including LBCLs and subtypes, indolent BCLs (including follicular lymphomas (FL), marginal zone lymphomas (MZL), and mantle cell lymphomas (MCL)) at Hôpital Maisonneuve-Rosemont, a 725-bed tertiary care hospital accredited by the Foundation for Accreditation of Cellular Therapy. Patients aged over 18 years who underwent alloHCT between 1 January 2015 and 31 December 2023 were included.

At our institution, alloHCT is recommended in first remission only for Richter’s transformation (RT). For other LBCLs, the indication is discussed on a case-by-case basis in the relapsed setting. Tandem auto-alloHCT is considered for LBCLs of indolent origin if the indolent component was previously treated. Tandem auto-alloHCT consisted of an autoHCT performed using BEAM (BCNU, etoposide, cytarabine, melphalan), BendaEAM (bendamustine, etoposide, cytarabine, melphalan), BEAC (BCNU, etoposide, cytarabine, cyclophosphamide), or MelVP16 (melphalan, etoposide) conditioning, followed 3–4 months later by a nonmyeloablative alloHCT with fludarabine and cyclophosphamide [1,2]. This strategy is based on a former publication which showed an overall survival (OS) and progression-free survival (PFS) of 96% at 3 years [3]. For indolent B-cell lymphomas, alloHCT is considered in ≥3rd line of therapy, with tandem auto-allo transplant for patients with co-morbidities prohibitive for myeloablative alloHCT. For MCL, alloHCT is offered for relapsed/refractory disease and discussed on a per case basis.

Additional criteria for alloHCT in our institution are the following: age ≤ 75 years, ECOG performance status of 0–1, Karnofsky Performance Scale (KPS) ≥ 70, acceptable age-adjusted Hematopoietic Cell Transplantation-specific Comorbidity Index (HCT-CI) score (5 if age ≤ 60 years, 3 if aged 61–64 years, 2 if ≥65 years, 0 if age 70–75) and cardiac ejection fraction ≥ 50% for myeloablative conditioning regimens and >30% for non-myeloablative conditioning regimens. A DLCO ≥ 50% corrected by the Dinakara method and an estimated glomerular filtration rate ≥ 60 mL/min are also required.

Peripheral blood stem cell grafts are preferred for both related and unrelated donors, with a target cell dose of 5 × 10^6^ CD34+ cells/kg and a minimum of 2.0 × 10^6^ CD34+ cells/kg. HLA-compatible grafts are always prioritized. Donor types are preferred in the following order: (1) matched related donors under 50 years of age; (2) matched unrelated donors, ideally under 40 years of age; and (3) haploidentical donors (preferably under 35 years) (4) HLA-mismatched unrelated donors with preference for class II mismatch and cord blood are considered on a case-by-case basis.

Our conditioning regimens are based on our institutional transplant protocol, which follows current international recommendations and is regularly updated based on the latest evidence to meet FACT accreditation requirements. Myeloablative conditioning was used for aggressive histologies, whereas indolent lymphomas received reduced intensity. Patients in need of debulking not immediately fit for alloHCT were selected for a tandem auto-allo approach.

GVHD prophylaxis included a calcineurin inhibitor (cyclosporine for related donors, tacrolimus for others) combined with either methotrexate or mycophenolate mofetil [4]. Our institutional standard for in vivo T-cell depletion is rabbit anti-thymocyte globulin (ATG) in HLA matched related donor and HLA matched unrelated donor, and post-transplant cyclophosphamide (PTCy) for one antigen HLA-mismatched donors and for haploidentical transplants.

The primary endpoints were OS and PFS. Secondary endpoints included NRM and incidences of acute and chronic GVHD. GVHD grading followed published guidelines from the MAGIC and NIH criteria [5,6]. Response assessment was performed using the Lugano criteria via CT or PET scans, depending on the modality used at time of transplant [7]. Primary refractoriness was defined as stable disease or progressive disease within one year of ending frontline treatment. Patients with RT who developed chronic lymphocytic leukemia after alloHCT without evidence of high-grade disease were considered in persistent remission for PFS evaluation.

Descriptive statistics were applied to collected variables. For survival analysis, the Kaplan–Meier estimator was used, as well as the cumulative incidence function for competing risks data. We did not compare alloHCT efficacy between BCL subgroups, as they have very different disease biology and results could be misleading. This study was approved by the Ethics committee of Maisonneuve-Rosemont Hospital.

## 3. Results

### 3.1. Patients

From 1 January 2015 to 31 December 2023, 55 patients with BCLs were treated with alloHCT at Maisonneuve-Rosemont Hospital. Table 1 describes patients’ characteristics. BCLs were classified into three distinct groups: LBCLs (*n* = 25), indolent BCLs (*n* = 16) and MCL (*n* = 14). LBCL subtypes were as follows: RT (*n* = 10), transformed FL (tFL, *n* = 9), DLBCL-NOS (*n* = 3), primary mediastinal B-cell lymphoma (*n* = 1), high-grade B-cell lymphoma with double rearrangement of MYC and BCL2 (*n* = 1) and DLBCL-ALK-positive (*n* = 1). Among indolent BCLs, 15 (93.7%) were FLs and 1 (6.3%) was MZL.

At time of transplant, median age was 57 years (32–70 years), and 35 patients (63.6%) were males. Patients had a median Karnofsky Performance Scale of 90 and median HCT-CI score of 1. Our patients’ population was enriched with adverse prognostic factors, including 7 (44%) of indolent BCLs with POD24, and 6 (43%) of MCLs with blastoid/pleomorphic morphology or P53/TP53 expression/mutation.

The median number of treatment lines before alloHCT was 2, 3, and 2 for LBCLs, indolent BCLs, and MCL, respectively, with 15 (27%) of patients who had previously received radiation therapy and 33 (60%) a prior autoHCT. Among the LBCL and indolent BCL patients who had a prior autoHCT, 10 had an auto-alloHCT tandem.

Except for patients with RT directed to alloHCT in first consolidation, the majority of our cohort had previously been treated with autoHCT (LBCLs: 10/15 (67%), FL: 12/16 (75%), MCL: 11/14 (79%)). No patient had received CAR-T or bispecific antibodies prior to transplant. Most patients had chemosensitive disease, with 49 (89%) patients showing an objective response (PR: 20%; CR: 69%) to salvage therapy before transplant.

Donor types were as follows: 11 (20%) received a graft from an HLA-matched (8/8) sibling, 32 (58%) from an HLA-matched (8/8) unrelated donor, 4 (7%) from a haploidentical donor, and 8 (14%) from umbilical cord blood. All patients received a calcineurin inhibitor (tacrolimus 82%, cyclosporin 18%), and methotrexate (40%) or mycophenolate mofetil (60%) as GVHD prophylaxis. In vivo T-cell depletion with rabbit ATG was used in 22 (44%) patients and PTCy for the 4 (7%) patients (all for haploidentical alloHCT). Nonmyeloablative conditioning regimens were used more frequently in indolent BCLs (*n* = 10; 63%) compared to LBCLs (*n* = 7; 28%) or MCLs (*n* = 4; 29%), due to the use of a planned tandem auto-alloHCT in indolent BCLs (44%). Myeloablative conditioning was used in 9 (36%), 2 (13%), and 4 (29%) patients, and reduced-intensity conditioning in 9 (36%), 4 (25%), and 6 (42%) patients with LBCLs, indolent BCLs, and MCLs, respectively. Conditioning regimen included radiation therapy for 19 (35%) patients. All grafts were peripheral blood stem cells or cord stem cells. The median number of CD34+ cells infused was 7.9 × 10^6^ for non-cord blood grafts and 2.2 × 10^5^ for cord blood transplants. All transplants were performed in isolated rooms with HEPA filters with standard PCP, anti-viral and anti-fungal prophylaxis. All patients were treated and followed at our site by the same transplantation team until relapse or a minimum of 5 years, whichever occurred first.

### 3.2. Efficacy

PFS and OS are presented in Figure 1 and Figure 2. Following alloHCT, the overall 5-year PFS and OS of the entire cohort were 59.6% (IC 95%: 44.8–71.7) and 73.9% (IC 95%: 58.9–84.1). After a median follow-up of 6.1, 5.8, and 2.4 years for LBCLs, indolent BCLs, and MCLs, 5-year PFS were 57.2% (IC 95%: 34.2–74.7), 81.2% (IC 95%: 52.5–93.5) and 39.0% (IC 95%: 14.3–63.3), respectively. The median PFS was not reached (NR) for LBCLs and indolent BCLs. The median PFS for MCLs was 1.72 years. The 5-year OS was 62.8% (IC 95%: 37.9–80.0), 93.8% (IC 95%: 63.2–99.1) and 68.1% (IC 95%: 35.4–86.8) for LBCLs, indolent BCLs and MCLs. Median OS was not reached for all BCLs.

Overall, 15 patients relapsed: 5 LBCLs, 2 FLs, and 8 MCLs. Among RTs and tFLs, only one patient with RT had a relapse of the aggressive lymphoma. Three MCL patients had pleomorphic or blastoid morphology (one of whom also had a TP53 mutation), and all relapsed. All MCL relapses occurred within the first 3 years after transplant.

At the time of alloHCT, six patients (5 RT and 1 FL) had progressive disease based on PET or CT scans. Two received a myeloablative conditioning and four received a reduced intensity. One received post-transplant radiation therapy consolidation. Interestingly, only three RT patients with progressive disease at time of alloHCT experienced a relapse: two with chronic lymphocytic leukemia (not included in the PFS curve) without evidence of RT and one with RT. All were treated with Bruton tyrosine kinase inhibitors and obtained good disease control. The other three patients who had progressive disease at time of alloHCT and achieved remission after transplant had chronic GVHD (two mild, one moderate).

### 3.3. Safety

The overall 5-year NRM was 16.9% (IC 95%: 8.2–28.3%), with a relapse incidence of 23.4% (IC 95%: 12.7–36.0%). In the subgroup analysis, the 5-year NRM for LBCLs, indolent BCLs, and MCLs were 24.5% (IC 95%: 9.6–42.9), 6.2% (IC 95%: 0.4–25.6) and 14.9% (IC 95%: 0.2–39.7) (Figure 3). The cumulative incidence of relapse was 18.4% (IC 95%: 5.1–38.0), 12.5% (IC 95%: 1.9–33.6) and 46.1% (IC 95%: 17.6–70.8) for LBCLs, indolent BCLs and MCLs, respectively (Figure 4).

Acute GVHD occurred in 43.6% of our cohort, with grade III-IV acute GVHD in 18.1%, no grade IV acute GVHD was observed. Among patients with acute GVHD, 16% were steroid refractory. At 3 years, overall and moderate/severe chronic GVHD incidence of were 49% and 34.5%. Among patients developing chronic GVHD, 12.7% were treated with ruxolitinib.

## 4. Discussion

We report the results of a contemporary cohort of patients with BCLs treated with alloHCT between 2015 and 2023. Our study has several limitations. First, in addition to being retrospective, the descriptive nature of the study prevents direct comparison of alloHCT efficacy with other cellular therapies, such as CAR-T cells or bispecific antibodies. Second, the overall number of patients is relatively small, and our cohort is heterogeneous in terms of disease. Among indolent lymphomas, all were follicular lymphomas except for one patient. Since the efficacy of alloHCT differs between follicular lymphomas and marginal zone lymphomas in retrospective registries [8,9], our results cannot be extrapolated to marginal zone lymphomas. Third, disease risk stratification could not be thoroughly assessed in our study. Prognostic scores were unavailable for most patients, as they were commonly referred from community centers. P53 expression and TP53 mutations in MCLs were assessed in only a minority of our patients, as this became routine at our center only in recent years. Similarly, molecular characterization of RT and tFL was not available at our center at time of disease diagnosis; thus, clonality between the aggressive and the indolent entity could not be assessed. Some patients may have had de novo DLBCL rather than true clonally related DLBCL and may have been overtreated. However, we believe the impact of this possibility should be limited, as the emergence of true clonally related RT is much more common than de novo DLBCL in patients with CLL [10].

The PFS and OS observed for LBCLs contrast with what has been previously reported. A large Center for International Blood and Marrow Transplant Research (CIBMTR) cohort identified 503 patients with DLBCL who underwent alloHCT between 2000 and 2012 after disease progression following autoHCT. The authors reported a 3-year probability of PFS and OS of 31% and 27%, with an NRM of 30% [11]. Among possible explanations for our different results, 19 of our 25 patients with LBCL had a disease of indolent origin (tFL, RT) rather than de novo LBCL, since the graft-versus-lymphoma effect is known to be more effective in indolent diseases [12,13,14]. However, it is important to highlight that, while the evidence supporting this view is intriguing, it remains inconclusive in the literature, and this manuscript does not aim to definitively resolve the question.

Regarding RT, alloHCT is recommended as consolidation by many guidelines if remission is achieved after salvage chemotherapy. Kim et al. recently reported on 28 patients with RT who underwent alloHCT between 2010 and 2019. Ninety-two percent of patients were chemosensitive at time of alloHCT, with a reported 4-year PFS of 53% and NRM of 32% [15]. Of concern, patients with RT undergoing alloHCT may relapse with the CLL component alone, without evidence of aggressive disease, as recently reported in a large cohort of 66 RT patients by the EBMT Malignancy Working party [16].

Transformed disease in a patient with previously treated FL is sometime referred for consolidation with an autoHCT. AlloHCT in this context is reserved for investigational purposes, and no recent prospective trial comparing alloHCT to autoHCT has been performed. Villa et al. retrospectively reported 22 patients treated with consolidation alloHCT and 97 patients with autoHCT. Compared to a historical cohort not undergoing transplantation, consolidation with alloHCT seemed to provide better PFS, whereas consolidation with autoHCT yielded better PFS and OS. When comparing allo to autoHCT, similar OS (46% vs. 65%, *p* = 0.24) and PFS (46% vs. 55%, *p* = 0.12) were observed; as expected, alloHCT had a higher NRM (23% vs. 5%) and was not recommended [17]. In the pre-rituximab era, a prospective alloHCT was associated with a decreased probability of disease progression (69% vs. 20%, *p* = 0.001) and may overcome the adverse risk factor of transformed disease [18]. Whether this can also be applied to patients treated in the rituximab area remains unclear. For patients with RT and tFL, our data suggest that allo transplant can achieve excellent long term disease control. Strict selection may also explain our results; our patients had good performance status, few comorbidities, and most had chemosensitive disease. Finally, 20% of our patients were approached with a tandem nonmyeloablative auto-alloHCT. This approach is still investigational and not considered standard of care, which limits its broad applicability. Yet, it has shown promising outcomes for high-risk lymphomas [1,3].

Our impressive results for indolent BCLs (mostly FL) correlate with the previous literature. The place of alloHCT remains unclear for indolent BCLs, but a graft-vs-lymphoma effect seems to be particularly strong, as several reports have documented lymphoma regression with the use of donor lymphocyte infusions or withdrawal of immunosuppression [12,13,19]. Sureda et al. reported more than 1500 patients from CIBMTR and the European Bone Marrow Transplant (EBMT) registries with relapsed/refractory FLs that underwent alloHCT from 2001 to 2011. In line with our own results, the 5-year median PFS and OS were 61 and 52%, respectively, with a 5-year NRM of 19% [9].

We report that a lower PFS for MCLs compared to other BCLs and alloHCT did not seem to overcome the impact of high-risk disease features, since 75% of MCL with P53/TP53 expression/mutation and all patients with blastoid or pleomorphic morphology relapsed within 3 years. These results contrast with previous reports, although our numbers are small. Retrospective analysis of alloHCT for MCL performed between 1999 and 2008 demonstrated a 5-year PFS and OS ranging between 30–50% and 40–60%, respectively [20,21]. For patients with P53 overexpression or TP53 mutation, alloHCT may overcome the negative prognostic impact of these alterations in some series [22,23]. However, NRM remains high, ranging from 20 to 35%, thus limiting its use [20,21,22,23]. Our data however suggests that alloHCT may not be the best therapy for these patients, more contemporary cohorts are needed and clinical trials should be prioritized.

The NRM observed in our population (16.9%) is lower than what has been described in previous reports, as most studies assessing alloHCT in BCLs were conducted before 2010. Efficacy and safety associated with alloHCT have improved over the past decades, starting with the use of high-resolution molecular HLA-typing. The incidence of both acute and chronic GVHD has decreased with the use of ATG and PTCy [24,25]. Additionally, the treatment landscape for both acute and chronic GVHD has changed dramatically with novel agents such as ibrutinib, ruxolitinib or belumosudil, thereby decreasing the morbidity and mortality associated with GVHD [26,27,28,29]. Infectious prophylaxis has also improved, with better CMV prophylaxis and treatment, better anti-fungal agents, and improved antibiotic selection. Conditioning has been refined and individualized to patients and disease characteristics [30]. As a result, outcomes following alloHCT have improved in recent years [31,32,33,34,35]. Importantly, the 5-year overall NRM observed in our population is comparable to the CAR-T-related NRM reported by some registries [36,37]. Although alloHCT and CAR-T cell therapy target different patient populations, it is reassuring to see that NRM has decreased with alloHCT in recent years and emphasizes that this therapy can be used safely in selected patients.

Whether alloHCT should be reserved for relapsing/refractory BCLs after CAR-T cell therapy remains uncertain. There is no comparative trial evaluating CAR-Ts vs. alloHCT for chemosensitive BCLs. An indirect comparison has been performed by Mussetti et al., who have reported similar PFS and OS for chemosensitive LBCLs, but higher NRM for alloHCT [38]. Recently, Zurko et al. have evaluated the outcomes of relapsed/refractory LBCL patients following CAR-Ts and treated with alloHCT, with a reported 1-year PFS of 45% and OS of 59% [39]. However, a significant NRM of 22% at one year was observed, confirmed by Shadman et al. (33% at one year, all B cell neoplasms included) [40], questioning whether alloHCT should be offered sooner for young and fit patients as it remains the only known potential curative option for several BCLs.

## 5. Conclusions

Our data support the role of alloHCT for BCLs with particularly high and prolonged response rates in FL and LBCLs of indolent origin (mostly RT and tFL), where the role of alloHCT is being challenged with new cellular immunotherapies. Although NRM remains a major concern in alloHCT, our real-world results demonstrate an acceptable 5-year NRM of 16.9% with modern transplants. In an era of increasing number of effective therapies, we believe it is imperative to personalize care, choose the most effective cellular therapy for patients and the best timing during the course of their illness. AlloHCT, being a potentially curative option, should be considered for fit patients with chemosensitive FL or LBCLs of indolent origin and a low comorbidity index.

## Figures and Tables

**Figure 1 curroncol-32-00285-f001:**
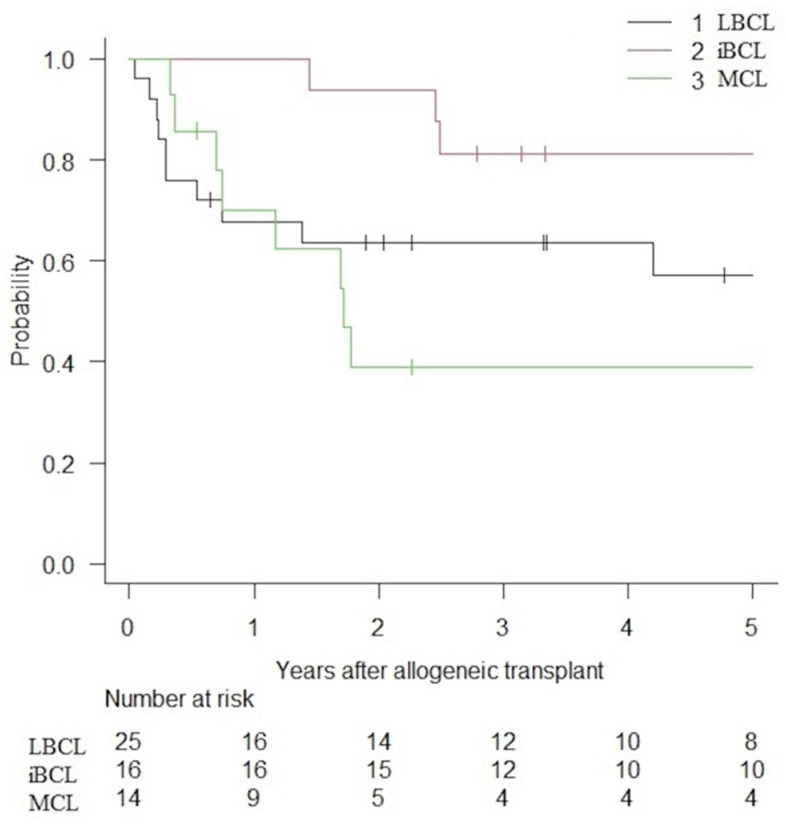
PFS of patients with BCLs treated with alloHCT.

**Figure 2 curroncol-32-00285-f002:**
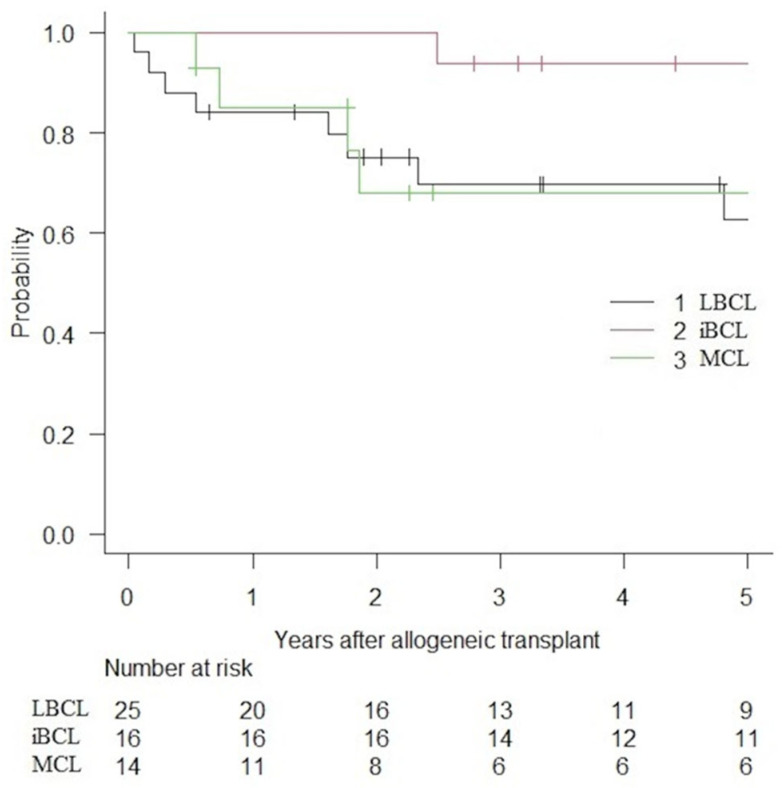
OS of patient with BCLs treated with alloHCT.

**Figure 3 curroncol-32-00285-f003:**
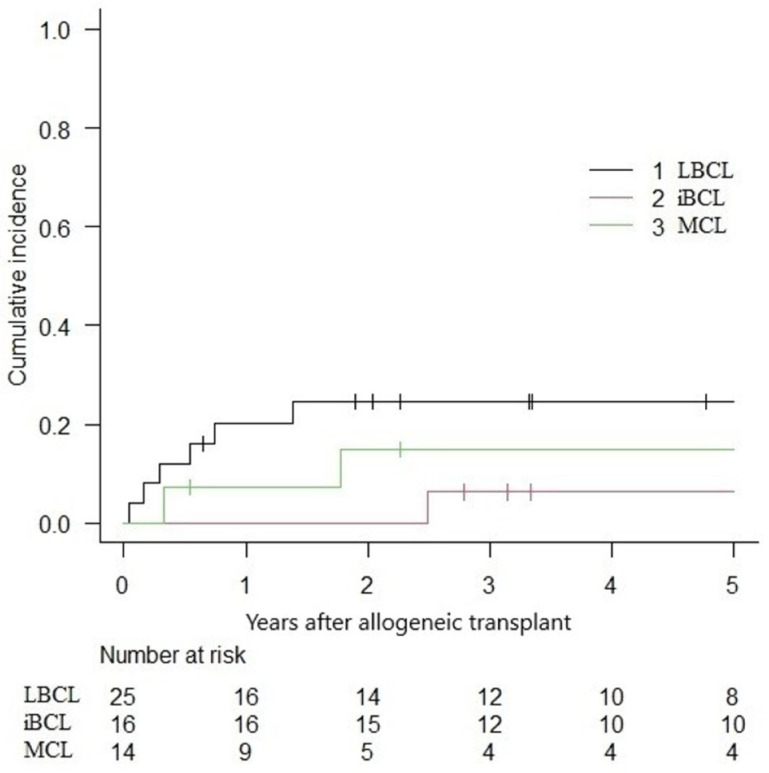
Non-relapse mortality following alloHCT.

**Figure 4 curroncol-32-00285-f004:**
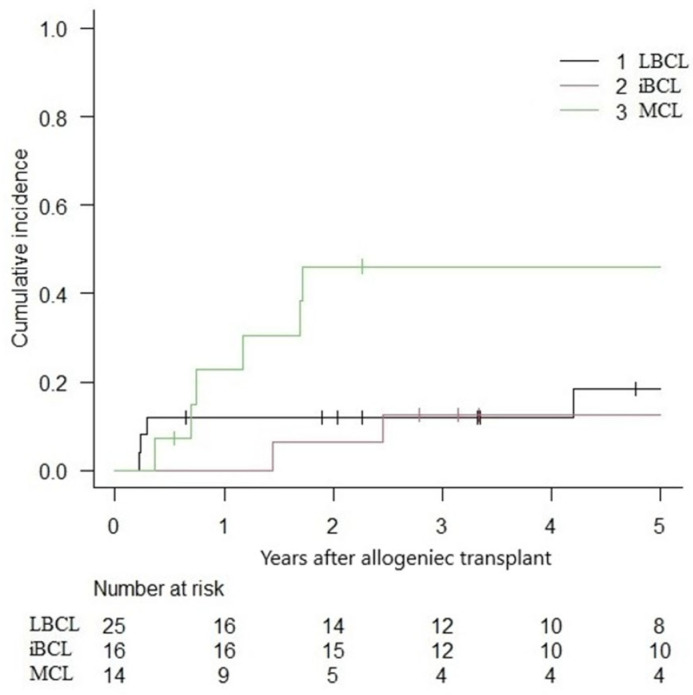
Cumulative incidence of relapse following alloHCT.

**Table 1 curroncol-32-00285-t001:** Characteristics of patients with BCLs treated with alloHCT between 2015 and 2023.

	LBCL *n* = 25	Indolent BCL *n* = 16	MCL *n* = 14
Subtypes	RT ^1^: 10 tFL ^2^: 9 DLBCL-NOS: 3 PMBCL ^3^: 1 DLBCL-ALK+: 1 HGBCL-DR ^4^: 1	FL: 15 MZL: 1	Blastoid/pleomorphic morphology: 3 P53/TP53 expression/mutation: 4/6 available
Age in years (range)	55 (28–68)	56 (30–70)	59 (46–67)
Male sex (%)	13 (52)	10 (63)	12 (86)
KPS ^5^	90	90	90
HCT-CI (range) ^6^	2 (0–6)	1 (0–3)	0 (0–3)
Treatment			
Median number of lines (range)	2 (1–6)	3 (2–7)	2 (1–3)
POD24 (%)	-	7 (44)	-
Primary refractory (%)	5 (20)	4 (25)	2 (14)
Radiation therapy (%)	7 (28)	4 (25)	4 (29)
AutoHCT (%)	10 (40)	12 (75)	11 (79)
Auto-alloHCT tandem (%)	3 (12)	7 (44)	0
Disease status prior alloHCT (%)			
Complete response	17 (68)	8 (50)	13 (93)
Partial response	3 (12)	7 (44)	1 (7)
Stable disease	0	0	0
Progressive disease	5 (20)	1 (6)	0
Type of donor (%)			
Sibling	4 (16)	3 (19)	4 (29)
Unrelated	12 (48)	11 (69)	9 (64)
Haploidentical	3 (12)	0	1 (7)
Cord blood	6 (24)	2 (13)	0
ATG ^7^ use (%)	9 (36)	5 (31)	8 (57)
PTCy ^8^ (%)	3 (12)	0	1 (7)
Conditioning (%)			
Myeloablative	9 (36)	2 (13)	4 (29)
Reduced intensity	9 (36)	4 (25)	6 (43)
Nonmyeloablative	7 (28)	10 (63)	4 (29)

^1^ RT: Richter transformation; ^2^ tFL: transformed follicular lymphoma; ^3^ PMBCL: primary mediastinal B-cell lymphoma; ^4^ HGBCL-DR: high-grade B-cell lymphoma with double rearrangement of MYC and BCL2; ^5^ KPS: Karnofsky Performance Scale; ^6^ HCT-CI: Hematopoietic Cell Transplantation-specific Comorbidity Index; ^7^ ATG: anti-thymocyte globulin; ^8^ PTCy: post-transplant cyclophosphamide.

## Data Availability

The data presented in this study are available on request from the corresponding author.

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
