# Peer review of "Allogeneic Hematopoietic Cell Transplant for B-Cell Lymphomas in the Era of Novel Cellular Therapies: Experience from a Tertiary Canadian Center"

_curroncol, 2025, doi:10.3390/curroncol32050285_

Round 1
Reviewer 1 Report
Comments and Suggestions for Authors
The authors have put together a compilation of their experience with allogenic stem cell transplantation (allo HSCT) for a variety of relapsed/refractory B cell lymphomas. However, it seems to lack focus. The final theme that the authors strive to drive at seems to be that their limited number patient data supports the role of allo HSCT for indolent B cell lymphomas and large B cell lymphomas of indolent origin and that given the relatively low NRM rates from their data set, allo HSCT should also be considered in the therapies offered to fit patients with chemo-sensitive disease of indolent origin.
Even though the title explores the role of allo HSCT for B Cell Lymphomas in the era of novel cellular therapies, the narrative takes seems to lack direction for a serious reader. In the introductory paragraph, the authors say the aim of the retrospective study was to examine the outcome of allotransplant to examine toxicity and identify which patients with B cell lymphomas (BCLs) could specifically benefit from this therapy.
In the materials and methods sections, they include 55 patients who underwent allo HSCT between 2015 and 2023 for all Large B Cell Lymphomas including subtypes, Follicular Lymphoma, Marginal Zone Lymphoma and Mantle Cell Lymphoma. In table 1, they include LBCL, Indolent BCL and MCL. Several areas could improve on more detail for example how many of the LBCLs were de novo large B cell lymphomas versus how many were transformed from indolent lymphomas is not very clear.
Re-structuring the manuscript with more granular detail and thematic focus and flow will help improve the appeal.
Reviewer 2 Report
Comments and Suggestions for Authors
The authors present their experience on allogeneic HSCT, in a monocentric, retrospective series of BCLs. The series has limited size and has been collected over a rather long time. Moreover, no policy in timing of the procedure and choice of conditioning regimen is reported. Interestingly, large cell lymphomas deriving from an indolent disease, account for the majority of LBCL and follicular lymphomas for all but one indolent lymphomas. Moreover, the authors do not present statistically significant findings deriving from their analysis.
As a general remark, the authors should improve their presentation by more thoroughly discussing the limitations of their series. The list of limitations in the last paragraph of the discussion is somewhat out of place; I feel it as foreign body rather than the start of a proper discussion. On the other hand, they consider some peculiarity of the series, as the composition of the LBCL group, but they overshadow the even more outstanding presence of 15/16 FL among indolent lymphomas.
Moreover, I am listing above minor points deserving some comment.
Summary
Acute GVHD occurred in 43.6% (18.1%)….chronic GVHD was 49% (34.5%). In this form, I do not understand. Likely, it derives from the questionable shrinkage of the corresponding, plainer sentence in the results section.
M&M
Likely, FL and MZL were collected under the heading indolent lymphomas; nevertheless, the term “indolent lymphomas” should be defined in this section, rather than in the Results.
The authors do not mention any method of statistical comparison between the possibly observed intergroup differences.
Conversely, they plan to perform a multivariate analysis, but they report none.
Results
The authors report in the text, percentages without absolute numbers, referred to the whole series. This makes reading uneasy and rather disturbing.
- The number of patients previously undergoing autologous HSCT, is different in Table 1 and in the text (in brackets).
As far as I understand, 10 patients received allogeneic HSCT as part of a tandem program. In my opinion, the matter should be clarified.
The criteria for delivering both ATG and post-transplant CTX, should be clarified, even if the only criterion was clinical judgement.
All relapses occurred within the first two years… By examining the PFS curve, at least one relapse seems to have occurred a bit later.
Only three RT patients with progressive disease…Two of the relapses were CLL; in M&M, the authors state that RT patients relapsing as CLL were not included in the efficacy analysis. Likely, they did not consider them by calculating PFS curves. In any case, the information is interesting. In my opinion, the authors should again clarify.
If no statistically significant difference was found among the presented curves, the authors should specify.
Discussion
158 The statement is interesting but questionably supported by the presented data. The authors should clarify.
208 The choice of conditioning regimens and the timing of allogeneic HSCT, possibly after CAR-T failure, are key points. Unfortunately, the authors cannot present any data contributing to the discussion, because they did not define any policy to choose conditioning regimen and no patient had previously received CAR-T in their series. It is legitimate for the authors to confront with these items; nevertheless, they should state that they are speculating rather than discussing their data.
- The sentence is too concise, in comparison to the relevance of the point.
1) In spite of some overlapping, indolent BCL and LBCL of indolent origin present differences as far as alternative therapeutic options are concerned.
2) The presence of 15/16 FL among indolent BCLs, likely positively affected the outcome of the group. Relying on the presented data, allogeneic HSCT could be performed sooner in FL rather than in indolent BCLs as a group.
3) On the other hand, letting alone the limited number of patients, there is also a difference between tFL and RT. In tFL, allogeneic HSCT may overcome adverse risk factor and lead to a long-term disease control, whereas CLL relapse is still a matter of concern in RT.
The authors are not expected to share my trivial consideration. They should more thoroughly discuss this key point, possibly at the expense of other paragraphs, if a matter of space is involved.
- Listing limitations of the series in the last paragraph of the discussion, is not an optimal choice, at least in my opinion. Limitations should be a starting point for discussion rather than a final remark.
Round 2
Reviewer 1 Report
Comments and Suggestions for Authors
Authors have addressed most of the concerns raised
Author Response
Thank you for reviewing our manuscript
Reviewer 2 Report
Comments and Suggestions for Authors
The authors did considerable effort in order to improve their manuscript. Few misunderstandings are left, likely engendered to my poor English stile.
- I agree with the authors that a descriptive analysis was appropriate for the purpose of the paper. Nevertheless, the last paragraph in M&M mentioned “statistical tests” and multivariate analysis. The authors erased multivariate analysis but on line 99 of the reviewed manuscript, we still read: “appropriate statistical tests were used”. This sentence should also be erased.
- I did not expect that any patient simultaneously received ATG and PTCy; my comment was not well written and eventually misleading, since I meant either ATG or PTCy. Nevertheless, the authors state, even in the revised form, that four patients received PTCy because of haploidentical HSCT, whereas Table 1 reports four patients receiving haploidentical HSCT and five patients receiving PTCy. On the other hand, the number of patients receiving ATG does not correspond to any obvious combination of subgroups. In my opinion, the authors should specify which criterion (if any) was used to deliver ATG.
- I agree with the disagreement of the authors. My comment was unclear. I share the authors view that LBCL deriving from indolent lymphomas, could do better the de novo ones after allogeneic HSCT. I simply liked to say that the evidence favoring this view is interesting but not conclusive in the literature, and that the present manuscript is not aimed at definitively addressing the question.
